# The Continuum of Microbial Ecosystems along the Female Reproductive Tract: Implications for Health and Fertility

**DOI:** 10.3390/pathogens11111244

**Published:** 2022-10-27

**Authors:** Claudia Nakama, Brice Thompson, Cory Szybala, Andrea McBeth, Piper Dobner, Heather Zwickey

**Affiliations:** 1Helfgott Research Institute, National University of Natural Medicine, Portland, OR 97201, USA; 2Thaena, Inc., Vancouver, WA 98661, USA

**Keywords:** lactic acid, reproductive immunity, unexplained infertility, *Lactobacillus*, vaginal community state types, urogenital microbiome, dysbiosis, metabolomics, bacterial vaginosis

## Abstract

The microbial ecosystem of the female urogenital tract is composed of many niche microenvironments across multiple organ systems in the urinary and reproductive tract. It is complex and contains a variety of bacteria, archaea, viruses, yeast, and protozoa—Many of which are still unidentified or whose functionality is unknown. Unlike the gut microbiome, whose composition is relatively stable in the absence of external perturbations, the urogenital microbiome is constantly shifting in response to biological cycles such as hormonal fluctuations during menstruation. Microbial composition differs between women but the dominance of some microbial families, such as *Lactobacillaceae* and other lactic acid-producing bacteria, are shared. Research suggests that it is difficult to define a universal healthy urogenital microbiome and consequently map a path to recovery from disease due to dysbiosis. Due to its temporal shifts, the female urogenital microbiome offers a unique opportunity to examine the biological mechanisms that work to restore a microbiome to its baseline. Common functional disorders in women’s health are often difficult to diagnose and treat, are prone to recurrence, and can lead to subfertility or infertility. Knowledge of the interconnected microorganism communities along the continuum of the female reproductive tract could revolutionize the quality of women’s healthcare.

## 1. Introduction

*Microbiota* is defined as the sum of the microorganism populations in a particular environment. Next-generation sequencing techniques such as 16S ribosomal RNA sequencing have allowed us to further classify microbial DNA on different phylogenic levels within the context of their niche environment. The *microbiome* can be thought of as an entire ecosystem, extending beyond microbial taxa to include host and bacterial genomes along with environmental factors. All these factors contribute to the resulting functionality (or dysfunctionality) of the ecosystem, thereby affecting metabolic pathways designed to facilitate crosstalk between organ systems [1].

In 2008, the National Institutes of Health (NIH) funded the human microbiome project to profile the microbial composition at different bodily sites in healthy adults. The human microbiome project further funded demonstration projects that sought to assess whether changes in the microbiome could be related to human health and disease. They found that an altered microbiota was associated with many disorders, such as obesity, diabetes, irritable bowel disease (IBD), bacterial vaginosis (BV), and preterm labor [2].

It is widely recognized that intestinal dysbiosis is linked to a variety of health problems and disease states including functional abnormalities in the brain, heart, and musculoskeletal system [3]. Research has begun to focus on the correlation between metabolic activity of the gut with urogenital health. In fact, research is suggesting that niche environments throughout the body, from the oral microbiome to the gut [4] through to the genital microbiome [5] are all interconnected and influential on each other. As we will discuss, these interconnected ecosystems are in constant communication and work together to interface with the host’s immune, physiological, and metabolic systems.

Urinary tract infections (UTIs) will impact one out of every two women by the time they are 30, with 30% of those women experiencing a recurrence of infection within six months of antibiotic treatment [6]. Bacterial vaginosis (BV) is a highly prevalent disorder of the vaginal microbiota among women of reproductive age and thought to affect at least 30% of women globally [7]. BV has been confirmed to be associated with adverse gynecologic and obstetric outcomes, such as increased susceptibility to sexually transmitted infections [8], pelvic inflammatory disease [9], and preterm birth [10]. Prolonged unresolved or recurring infections with BV are thought to be associated with endometrial cancer pathogenesis [11] and infertility [12]. Infections of the genital tract, e.g., endometritis and pelvic inflammatory disease (PID) are caused by the ascension of pathogenic bacteria to the uterus [13], although the mechanisms are not fully understood.

Current statistics from the CDC indicate that 1 out of every 5 (or 20%) of women aged 15–49 experience clinical infertility (defined as trying unsuccessfully for 1 year) and 1 in every 4 women have trouble carrying a pregnancy to term [14]. The epidemic of subfertility and clinical infertility in the US is currently being treated without regard to lifestyle or intrinsic host factors such as hormonal and microbial balance. Instead, couples are referred to expensive and invasive assisted reproductive techniques such as in vitro fertilization (IVF) that do nothing to identify or address the underlying causes of subfertility. Furthermore, infection with pathogenic microorganisms that contribute to subfertility can lead to serious complications during pregnancy including premature delivery [15], miscarriage [16], intrauterine growth restriction [17], term stillbirth [18], preeclampsia [19], and other adverse outcomes. Therefore, investigations into immunological and microbial factors contributing to subfertility is a worthwhile scientific endeavor.

Studies using a combination of 16S rRNA amplicon sequencing and various culture techniques have identified distinct microbial microenvironments along the reproductive tract: in the cervical canal, uterus, fallopian tubes, peritoneal fluid, and vagina [20]. Together, these microbial environments represent a continuum which demonstrates an intra-individual shift in taxa and composition and varying community functionality. The homeostatic balance within and between these environments may possibly provide protection against disease as well as provide for optimal conditions for achieving and maintaining pregnancy. All of the environments along the reproductive tract undergo significant shifts in composition commensurate with biological factors such as monthly menses and transitioning into perimenopause [21]. Additionally, research has shown a bi-directional influence between the microbiomes of distal organ systems and the reproductive tract [22]; however, the downstream effects of dysbiosis seem to depend on individual immunological factors in the presence of external perturbations. The ability to define the relationship between hormonal and microbial changes as correlative versus causative remains elusive. This review discusses current understandings regarding the compositional characterizations and temporal dynamics of various microbial niches along the female reproductive tract. We will highlight the influence of neighboring environments on each other and how they interface with host immunological factors to either maintain homeostasis or contribute to the pathogenesis of disease. Finally, we discuss how recurrent disruptions in microbiome of the lower reproductive tract can lead to dysfunction in the upper reproductive tract and ultimately cause more complex functional complications such as infertility.

## 2. The Vaginal Microbiome: An Influencer of Microbial Niches along the Female Reproductive Tract

A “healthy” vaginal microbiome has been defined as being compositionally dominated by the family *Lactobacillaceae* [23]. Lactobacilli are well-adapted to the vagina and serve as a “first line of defense” against colonization by pathogens. Lactic acid is the main fermentation by-product of lactobacilli, and acts to keep the pH of the vagina low, ideally between 3.5-4.2. This pH range also helps to inhibit many other invading microbes [24]. Lactobacilli further protects the vaginal environment by producing bacteriocins which function as narrow-spectrum natural antibiotics by increasing the permeability of target cell membranes [25]. Moreover, lactobacilli elicit a stimulatory effect on our innate immune system by upregulating IL-23 thereby potentially activating the CD4^+^ T-helper subtype 17 (Th17) pathway [26]. *Lactobacillus* spp. undoubtedly confer many benefits to the vaginal microenvironment, however, observed immuno-modulatory effects suggest that the paradigm of a Lactobacilli-dominated vagina as the singular defining characteristic of vaginal health may be too simplistic.

Using high throughput sequencing models, researchers have identified 5 distinct community state types (CST) of vaginal microbiota [27]. CST I- III and V are dominated by lactobacilli species. CST I is dominated by *Lactobacillus crispatus*, CST II *Lactobacillus gasseri*, CST III *Lactobacillus iners*, and CST V *Lactobacillus jensenii*. CST IV is a mix of strict and facultative anaerobes: *Gardnerella*, *Atopobium*, *Mobiluncus*, and *Prevotella*. CST IV can be further broken down into two subtypes (CST IV-A and IV-B), of which only one, CST IV-B, is considered dysbiotic. Communities of state type IV-A were generally characterized by modest proportions of either *L. crispatus*, *L. iners*, or other *Lactobacillus* spp., along with low proportions of various species of strictly anaerobic bacteria such as *Anaerococcus, Corynebacterium, Finegoldia,* or *Streptococcus*. In contrast, communities of state type IV-B had higher proportions of the genus *Atopobium*, in addition to *Prevotella*, *Parvimonas*, *Sneathia*, *Gardnerella, Mobiluncus*, or *Peptoniphilus* and several other taxa [27]. While CST IV-B is exclusively associated with dysbiosis, and CST I is widely regarded as the “most stable” (possibly because *L. crispatus* maintains the lowest vaginal pH at 3.5) [28]. However, identifying a singular “healthiest” vaginal microbiome is more complex than considering taxa composition alone.

Women naturally shift between CSTs during the menstrual cycle and there have been observed patterns to these cyclical shifts. There are some ground states that have the propensity to shift towards a particular CST. For example, CST I will generally shift to CST III and then back to CST I, but almost never transitions to CST II or IV [29]. Many studies have sought to use CST characterization to understand vaginal microbiome dynamics and findings suggest that healthy subjects tend to persist in a CST for at least 2-3 weeks. More frequent shifts plus the presence of certain bacteria (e.g., *G. vaginalis* and *L. iners*) was a strong predictor of dysbiosis and resultant symptomatic disorders [29]. A longitudinal study by Gajer et al., examined the temporal dynamics of vaginal bacterial communities in healthy reproductive-age women. The communities of some women seemed more resilient showing predictable changes between community states that occurred with the menstruation cycle. However, the communities of other women were almost invariant during menses, and still others changed states continuously over time independent of menstruation. The ability of resilient community states to return to their “ground state” seems to rely on whether there is functional community redundancy particularly in lactic acid producing species [30].

Studies have shown that there are racial and ethnic variations in inter-species composition. Hispanic and African American women seem to favor CST IV-A, while Caucasians and Asian women tend to be *Lactobacillus* dominant, (CST I-III). Additionally, white women had more species diversity within their lactobacilli population, while women of color generally favored a singular *Lactobacillus* spp. [31,32]. The species-specific protective mechanisms also seemed to be racially determined. A case–control study by Elovitz et al., surveyed the cervicovaginal fluid of pregnant women to find that β-defensin 2 lowered the risk of preterm labor in African American women, while the reverse was true for all other races. Additionally, low β-defensin-2 was associated with increased risk of preterm labor (PTL) even in a vaginal environment rich in *Lactobacillus* spp. [33]. This further supports the theory that there are additional factors other than microbiota taxa acting to modulate immune response and therefore defining a healthy vaginal microbiome is difficult when considering all confounders.

As previously mentioned, bacterial communities in the vaginal ecosystem are not static, there are many disturbances through the efflux of cervico-vaginal fluid (CVF), antimicrobials, and sloughed epithelial cells. Community state types (CSTs) can also shift in response to cyclical changes in estrogen [34], glycogen content, pH, menses, as well as the introduction of exogenous bacteria due to human activities such as sex and hygiene practices [35] (See Table 1). A better understanding of factors that lead to the development and maintenance of specific and stable vaginal bacterial communities is needed so that strategies can be developed to promote and maintain reproductive health.

## 3. Intrinsic Host & Extrinsic Environmental Factors That Contribute to Vaginal Dysbiosis

Estrogen acts to raise *Lactobacillus* levels by increasing free glycogen availability in the vaginal mucosa. Extracellular glycogen levels peak during the follicular phase of the menstrual cycle and then decline just before ovulation. The fall of estrogen (and in turn glycogen production), coupled with increased iron and pH levels due to menstrual blood in the vaginal canal, results in a shift in community composition that potentially creates a window of vulnerability against invading pathogens [44]. Studies have shown that if there is community redundancy for lactic-acid production then the CST should be able to return to its “ground state” and maintain eubiosis [45,46]. In other words, metabolic activity must be preserved despite shifting species composition. Estrogen levels also drop after menopause; postmenopausal women have been found to be at an increased risk for UTI, bacterial vaginosis (BV), and other disorders rooted in dysbiosis [47,48].

The introduction of semen into the vagina provides complex issues for the vaginal microbiome. Seminal fluid induces mucosal changes along the female genital tract to increase the chance of pregnancy and contains several immunologically active molecules that both promote and inhibit female genital inflammation [49]. Females have developed an additional immunosuppressive mechanism to avoid reacting to semen: oxytocin release during orgasm. Studies have shown that oxytocin suppresses Th1 and Th17 responses [50,51], reducing the likelihood that women will have an inflammatory reaction to semen. However, this immunosuppression may leave women vulnerable to pathogenic microbes and the deleterious overgrowth of commensals possibly contributing to sexually transmitted infections.

Choice of contraception can also affect a woman’s microbial niche environments. While hormonal IUDs can alter estrogen levels to possibly exert a protective effect in the vaginal microbiome, oral hormonal birth control negatively contributes to the relationship between estrogen and the bacteria in the gut, altering gut permeability and immune inflammatory response [52]. Copper IUDs have a direct deleterious effect on the cervico-vaginal biome by increasing the colonization of BV associated microbiota. A parallel longitudinal cohort study by Achilles et.al., found that BV prevalence increased in women initiating copper intrauterine devices from 27% at baseline, 35% at 30 days, 40% at 90 days, and 49% at 180 days. Although the concentration of beneficial lactobacilli did not change over 6 months, there was a significant increase in the log concentration of *G. vaginalis* and *A. vaginae* [37].

## 4. Bacterial Vaginosis

Bacterial vaginosis (BV) is one of the most widespread disturbances in women of reproductive and menopausal age and is the most common diagnosis associated with the symptom profiles of malodor and discharge. This condition tends to be clinically overlooked due to a high degree of variability in the manifestation of symptom severity and persistence. BV is characterized by concurrent lowering of vaginal pH, loss or lowering of lactic acid producing species, and an increase of opportunistic anaerobic and facultative bacterial species: *G. vaginalis*, *Mobiluncis* spp., *Bacteroides* spp., *Peptostreptococcus* spp., *Mycoplasma hominis*, and *Ureaplasma urealyticum*. In research settings, BV is diagnosed using the Nugent scoring system [53], which utilizes a 0 to 10 score to estimate the presence of the three vaginal bacterial morphotypes that are characteristic of BV (*Lactobacillus*, *Gardnerella*, and curved gram rods) using Gram-staining and microscopic examination. While the Nugent system is quite accurate, its use is too cumbersome in clinical practice due to the high microscopy skill level required. The Amsel criteria, originally published in the American Journal of Medicine in 1983 [54], provides a more accessible way to clinically define the diagnosis of bacterial vaginosis using four criteria: the presence of excessive thin white vaginal discharge, a fishy unpleasant vaginal odor, an elevated vaginal pH (>4.5), and the presence of “clue” cells (squamous epithelial cells covered with adherent bacteria).

### 4.1. Bacterial Vaginosis and the Urinary Tract

Bacterial vaginosis co-presents with functional disorders in the urinary tract such as cystits, overactive bladder syndrome and UTI [55]. One study found an inverse relationship between urobiome diversity and symptomatic BV, correlating a lower diversity in the urinary biome with symptomatic pathogenesis of BV [56]. In a study by Gottshick et. al., the urinary biome was clustered into eight different urotypes. All urotypes were clustered according to the abundance of species, respectively: *Prevotella amnii*, *Sneathia amnii, Gardnerella vaginalis* and *Atopobium vaginae* (UT1), *Lactobacillus iners* (UT2), *Enterobacteriaceae* (UT3), *Enterococcus faecalis* (UT4), *Streptococcus agalactiae* (UT5), *Citrobacter murliniae* (UT6), and *Lactobacillus crispatus* (UT7). Interestingly, all urotypes except for UT7 were indicated in women with and without BV symptom expression [57]. This suggests that there is no singular composition profile that suggests a disease state, but rather hints towards a difference in innate host factors that could affect bacterial virulence factors and metabolic stress pathways. As such, the profiling of a “healthy” microbiome is difficult; metagenomic analysis has raised more questions about the interrelatedness of environments and community functionality.

### 4.2. Bacterial Vaginosis, Diseases of the Reproductive Tract, and Adverse Pregnancy Outcomes

Serious adverse reproductive health outcomes have been associated with BV, including infections of the upper reproductive tract such as salpingitis, [58] (inflammation of the fallopian tubes), adenomyosis [59] (inflammation of the uterus), and infections of the upper genital tract such as pelvic inflammatory disease [60] and endometritis [61]. BV has further shown an increased susceptibility to STIs, including chlamydia, gonorrhea, human papilloma virus (HPV), and human immunodeficiency virus (HIV) [62]. Furthermore, chronic, recurring BV may progress beyond the above-mentioned disease states and lead to an increased risk of infertility [63] and adverse pregnancy outcomes. Fichorova et al. [64] showed that BV organisms prevalent in the placenta of very preterm neonates, e.g., *Prevotella*, *Gardnerella*, anaerobic streptococci, peptostrepto-cocci, and genital mycoplasmas, were associated with the upregulation of systemic inflammatory mediators in the newborns. Additionally, low birth weight (LBW) is an important public health marker as LBW is the complication most associated with neonatal mortality. More than 80% of neonatal deaths are newborns with LBW, of which two thirds are born preterm and one third are born term but are small-for-gestational-age (SGA). BV has been associated with LBW and SGA [65].

### 4.3. Bacterial Vaginosis and the Pathogenesis of Tubal Infertility

The ciliated cells in the fallopian tubes are integral to gamete, sperm, and embryo transfer. The cessation of ciliary function, either through cellular death or the loss of the ciliary beat, is one of the causes of tubal infertility [66]. Infection by BV associated microbiota: *Mycoplasma hominis*, *Mobiluncus*, and *Bacteroides ureolyticus* have been shown to cause direct tubal damage or altered ciliary activity [67]. Species of *Mobiluncus* (*Mobiluncus curtisii and Mobiluncus mulieris*) produce cytotoxins that reduce ciliary activity and lipopolysaccharide endotoxins, released from *B. ureolyticus*, that damage the mucosa of the fallopian tube causing sloughing of cells and loss of ciliary activity [68]. In an organ culture study, fallopian tubes challenged with *G. vaginalis* caused the complete cessation of ciliary beat within 2 days after inoculation. Furthermore, vaginolysin (a human-specific pore-forming cytolytic exotoxin produced by *G. vaginalis*) is known to damage vaginal epithelial cells causing subclinical pelvic inflammation [69].

The recommended first-line therapy for the treatment of BV is metronidazole or clindamycin [70]. However, a study by Bradshaw et. al., found that after treatment with metronidazole, 58% of women had a recurrence of BV and 69% had a recurrence of abnormal vaginal flora after 12 months [71]. This highlights the need to improve the current approaches to BV treatment. While the exact causes of BV remain elusive [72], and there is contentious debate as to how it should be diagnosed, the condition is clearly a risk factor for adverse sequelae that affect women’s health. As we have discussed, it is possible for BV to progress to more serious disease states and cause irreparable harm to female fertility and is therefore a significant threat to women’s reproductive health, especially given its prevalence. Better knowledge of vaginal community dynamics and the mutualism between the host and indigenous bacterial communities is needed for the development of strategies to manage the vaginal ecosystem in a way that promotes functional health and minimizes the use of antibiotics.

## 5. The Microbiomes of the Fallopian Tubes, Endometrium, and Cervix

In general, the α-diversity of microbial taxa along the reproductive organ system decreases and lactobacilli concentrations increase as you travel downwards towards the vagina. Environments that are proximal to the gut (and therefore susceptible to bi-lateral spread from the gut) such as the fallopian tubes contain almost no *Lactobacillaceae* and are highly diverse [73]. Environments distal to the gut and closer to sources of external perturbations such as the vagina, posterior fornix, and cervix are lower in diversity and are *Lactobacillaceae* dominant (See Figure 1). The cervix and endometrium can be thought of as areas of transition that also exhibit predictable temporal shifts in composition commensurate with a variety of biological events. The uterus contains approximately 30% lactobacilli with moderate diversity and bacterial genuses such as *Pseudomonas*, *Acinetobacter, Vagococcus,* and *Sphingobium*. At the phylum level, Firmicutes dominated the lower reproductive tract while there were large proportions of *Proteobacteria, Actinobacteria*, and *Bacteroidetes* in the upper reproductive tract [74].

### Pathogenesis of Dysbiotic Disorders along the Upper Reproductive Tract

The microbiome of the uterus is of great interest as bacterial infection in this environment can have devastating effects upon achieving and maintaining pregnancy. A pilot study in 2016 showed that women with a non-*Lactobacillus* dominant uterine environment (defined as <90% *Lactobacillus* spp. in overall composition) experienced a 40% drop in pregnancy rates compared to women with a microbiome that was composed of >90% *Lactobacillus* spp. It was theorized that lactobacilli act to reduce inflammation in the endometrium, thereby improving implantation rates [75].

Despite being adjacent to the bacterially colonized vagina, it was previously thought that cervical mucous maintained uterine sterility. However, ascension of bacteria from the vagina is accomplished via a physiological uterine pump designed to pass semen through small channels in the cervical mucus [76]. Additionally, bacteria can be introduced into the uterus via hematogenous spread from transmembrane gut leakage into the peritoneal cavity and through retrograde ascension from the fallopian tubes [77]. Furthermore, one study found that *Enterococcus fecium* placed in the oral cavity of germ-free mice traveled to the placenta [78], suggesting that bacterial descension from the oral cavity is also possible.

Pelvic inflammatory disease (PID) and endometritis are upper genital tract infections with a range of clinical presentations and manifestations. The symptom profile can oftentimes be nonspecific tenderness at the cervix, or uterine and adnexal tenderness [79]. Symptoms of acute endometritis are similar to those of PID and, outside of pregnancy, the terms are often used interchangeably. Acute PID is caused by the ascension of strict or facultative anaerobes from the vagina to the endometrium lasting for fewer than 30 days [80] while chronic endometritis is an infection that lasts ≥30 days. PID and endometritis are associated with infertility through highly specific disorders such as ectopic pregnancy [81] and tubo-ovarian abscess [82]. Overall, the presence of BV-associated bacteria within the endometrium has been linked to a 3.4-fold increased risk of infertility [83]. A case–control study performed in England with women aged 16–45 found that greater than 85% of PID cases are caused by BV-related bacteria or STI’s, with fewer than half of those cases caused by STI associated bacteria such as *Neisseria gonorrhoeae* or *Chlamydia trachomatis*, suggesting an important role for ascension of BV-associated anaerobic bacteria and other non-BV-related pathogens in endometritis and PID pathophysiology [84].

Current research does not support the characterization of a normal or healthy uterine microbiome. However, it is possible that resident microbiota plays a role in pregnancy achievement through supporting implantation and placentation. Furthermore, the microbiome influences immunomodulatory factors that may influence the maturation of the fetal immune system. A mouse model study done by Thorburn, et.al. found that acetate, a molecule made by maternal gut microbes, crossed the placenta to the developing fetus. This maternally derived short chain fatty acid permanently altered naive T-cells in the fetal thymus skewing T-cell differentiation toward a regulatory phenotype, as opposed to an inflammatory one [85]. Moreover, epidemiological studies have tentatively linked maternal diet and antibiotic exposure to the development of neurodevelopmental disorders in children [86,87]. More research needs to be done to fully understand these functional pathways.

## 6. Immunological Responses, Redox Potential & Metabolic Function

The introduction of pathogens to the body stimulates an immune response. In the vagina, pathogenic viruses and bacteria bind to a TLR which then triggers Th1 and/or Th17 responses. Th1 responses are characterized by production of interferon gamma, CD8^+^ T cells, and macrophages; while Th17 responses involve IL-17A, IL-17F, and IL-22 [88]. This immune response is one way that overgrowth of pathogenic bacteria is controlled.

The environmental milieu of cytokines, chemokines, and bacteria in various organ sites affect T-cell differentiation [89]. The uterus relies on immunological factors that shut down inflammatory responses so that the developing blastocyst can implant into the uterine wall. More specifically, CD4^+^ regulatory T-cells (Treg) promote tolerance interactions between the endometrial lining and the embryo (i.e., apposition, successful implantation, and early placental morphogenesis) by down-regulating Th1 and Th17 immune responses [90]. Treg cells also inhibit further T-cell proliferation, suppress B-cell activation and antibody production, reduce natural killer cell cytotoxicity, and decrease proinflammatory cytokine production. Embryos have cytokine receptors from time of conception until implantation; alterations from the eubiotic uterine microbiome may affect the local cytokine/chemokine profile potentially altering embryonic development, gene expression, and viability. Additionally, *Bacteroides* (such as *Prevotella*) have been associated with an increase in Th1 cell differentiation, causing further inflammation [91].

Pathogenic bacteria lead macrophages and neutrophils to produce reactive oxygen species. These charged oxygen ions affect pH and redox potential in the vagina and uterus. Change in pH affects enzyme activity; many enzymes require a narrow pH window to function properly [92]. The microbial metabolism of anaerobic bacteria into organic acid pairs such as succinate/fumarate and lactate/pyruvate were found to contribute to the redox potential of the vaginal epithelium. Women with BV were found to have a low redox potential and therefore a more reduced vaginal environment than healthy women [93]. The presence of short chain fatty acids can influence vaginal redox potential and may indicate a reduced vaginal environment that favors the overgrowth of BV-associated anaerobes in addition to signaling immunological changes in the vaginal ecosystem contributing to BV pathogenesis [94]. The redox potential mechanism generally operates independent of other environmental conditions, further obscuring the complete profile of host factors that contribute to the pathogenesis of BV.

Previous studies have cited the production of lactic acid as a major conserved characteristic maintained through the functional redundancy exhibited by members of a transitioning community. While fluctuations in community composition are affected by identifiable extrinsic perturbations during windows of vulnerability, the length of time that an individual can maintain functionality when in a transitory CST remains unknown. Additionally, we are not able to identify a tipping point at which an individual would not be able to naturally restore their baseline CST. Vaginal communities of some women show low constancy and high levels of species turnover, and community dynamics can vary widely among women, even among those that cluster in the same community class. In total, 20–30% of reproductive-age women have diverse lactic acid-producing microbiota that are not lactobacilli, and vaginal microbiota can experience dramatic transitions in community composition over time without transitioning to dysbiosis [95,96]. Therefore, microbial metabolites could be useful biomarkers for BV, as they reflect changes in vaginal microbiota that are associated with BV but overcome the difficulty of attempting to link specific bacterial species/strains, with uncharacterized metabolic capacities that may be functionally redundant. Longitudinal study designs that incorporate metabolomic along with metagenomic analysis could offer a more profound comprehension of microbiological immunity. Along with identifying microorganisms, we could determine their metabolic capacity, antimicrobial and immune modulatory effects, and function within the microbiome.

## 7. Conclusions

Defining a “healthy” or eubiotic microbiome remains a challenge. While we see patterns of correlation between certain disease states and the presence or absence of certain bacterial species, the causative reasons for pathogenesis of disease and the subsequent factors that affect the ability or inability to return to homeostasis remains complex. Microbiota can have drastic effects on the urogenital immunity system—and therefore the clinical sequelae of pregnancy or the pathogenesis of infection—by a variety of mechanisms. Firstly, the microbiome influences inflammatory responses of the innate immune system through the regulation of signaling pathways. Secondly, microbiota can also act to alter the epithelial barrier integrity. The native microbiota acts to starve out pathogenic bacteria by evolving to become the best nutrient scavenger in their native environment—known as the “Competitive Exclusion” concept. Lastly, microbiota make metabolites, and certain metabolites can either suppress the growth of pathogenic bacteria or make an environment more susceptible to pathogenic bacterial overgrowth. While compositional classification remains a valuable way to categorize the microbiome, current observations suggest that microbial composition is highly individualized and only sometimes correlates with disease states, therefore, microbiome mediating strategies should also be highly individualized. Current longitudinal study designs do not fully explain the nature of shifting communal composition for the individual nor does it allow us to assess the preservation of community functionality, the knowledge of which would further elucidate the factors that contribute to dysbiotic states.

The health of the urogenital microbiome is dependent on the health of its individual niche environments and on several intrinsic and extrinsic host factors including pH, hormone levels, daily lifestyle activities, and more. To truly understand what defines health in the female urogenital microbiome we will have to look at many of the current hypotheses/scientific concepts via a holistic ecosystem approach.

## Figures and Tables

**Figure 1 pathogens-11-01244-f001:**
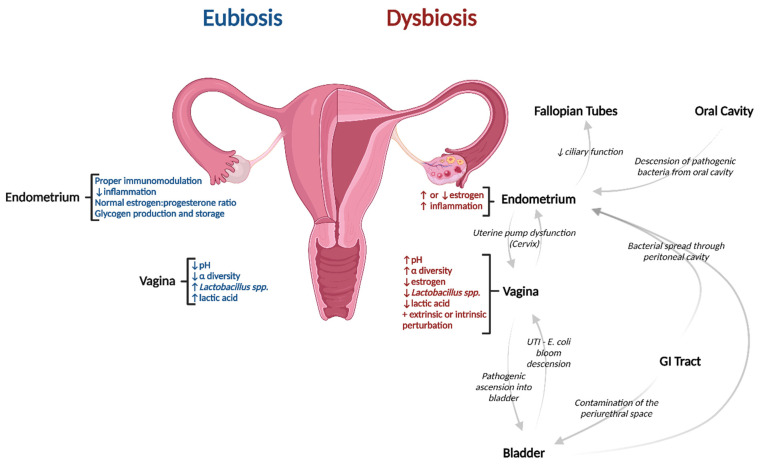
Characteristic differences between eubiosis and dysbiosis of the female urogenital microbiome and how dysbiosis can result from several interconnected regions. Created with BioRender.com. Abbreviation: UTI—Urinary tract infection.

**Table 1 pathogens-11-01244-t001:** Known disruptions to the urogenital microbiome. Abbreviations: spp.—species; IL—interleukin; IUD—intrauterine device; LNG—levonorgestrel; CST—community state type; GI—gastrointestinal; HPV—human papilloma virus.

Disruption	Mechanism	Result	Reference
**Biological**			
**Menstruation**	Temporary dip in estrogen during the secretory phase	Estrogen promotes proliferation of *Lactobacillus* spp. Responsible for maintaining a low vaginal pH. A decrease in estrogen levels results in a decrease in Lactobacilli and a temporary rise in vaginal pH which increases susceptibility to opportunistic pathogens.	Bardos et al., 2020 [36]
**Menopause**	Decrease in overall estrogen levels	Chronic low estrogen levels decrease accumulation of glycogen in the vaginal epithelium thereby reducing colonization of lactobacilli dependent on epithelial glycogen levels.	Achilles et al., 2018 [37]
**Pregnancy**	Inflammatory responses are moderated	Towards the end of pregnancy, the lower reproductive tissues increase microbial diversity and decrease *Lactobaccilus* spp.	Bardos et al., 2020 [36]
**Lifestyle**			
**Hygiene products**	Changes to pH levels	Vaginal douching has been linked to increased susceptibility to bacterial vaginosis.	Brotman et al., 2008 [38]
**Tampons**	Changes to lactic acid production and pH	Decreased vaginal microbiome stability is associated with tampon use.	Carter et al., 2018 [39]
**Antibiotic use**	Killing of commensal bacteria	Depletion of commensal bacteria due to antibiotic treatment was found to cause increased secretions of IL-33 in vaginal epithelium, suppressing antiviral immunity.	Oh et al., 2016 [40]
**IUD-use**	Changes in commensal microbial populations	Copper IUD use was associated with an increase in *Gardnerella vaginalis* and *Atopobium vaginae* possibly leading to increased bacterial vaginosis risk. LNG-releasing IUD use was associated with a decrease in *Lactobacillus* spp. Furthermore, an increase in *Candida* spp.	Achilles et al., 2018 [37]
**Sexual activity**	Changes in commensal microbial populations, shift in CST type	The number of sex partners and age when sexual activity started is associated with CST grouping and microbiome stability. Women with new sexual partners are at an increased risk for bacterial vaginosis.	Chen et al., 2017 [8]
**Causative Pathogenesis**			
**Leaky gut**	Hematogenous spread of GI commensals into peritoneal cavity	Increased risk of placental infection, pre-term birth, and miscarriage	Bardos et al., 2020 [36]
**Periodontal disease**	Microbial transmission from oral cavity	Increased risk of intra-uterine infection and pre-eclampsia	Parihar et al., 2015 [41]
**Resultant Pathogenesis**			
**HPV infection**	Opportunistic infection	Subjects with HPV infections have less *Lactobacillus* spp. Furthermore, more microbial diversity	Nicolò et al., 2021 [42]
**Cervical cancer**	Variable gene expression of *L. iners*-dependent community functionality and composition	*L. Iners* decreased amongst women with HPV infection, suggesting a protective effect. Alternatively, *L. iners* is associated with CIN 2+ and ICC	Piyathilake et al., 2016 [43]

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
