# Peer review of "The Continuum of Microbial Ecosystems along the Female Reproductive Tract: Implications for Health and Fertility"

_pathogens, 2022, doi:10.3390/pathogens11111244_

Round 1
Reviewer 1 Report
This article seeks to review the literature on the female urogential microbiome as it relates to pathogenesis. This is a large and broad topic. The article is split into sections with titles intending to focus on specific aspects of the topic. However, the organization of the material does not always match with the section headings and it is unclear how and why the authors decided on which subtopics to discuss. It is also not obvious what the authors are referring to with the term "recovery" in the title and where in the article they discuss recovery. Because of these organization issues and the lack of citations of the primary literature for numerous statements in the text, this review article is not useful in its current form. I have listed several examples of places in the article that need attention with respect to clarity, organization, or most importantly, proper citation of the literature. This list is not exhaustive and the whole article should be checked for sufficient citations of the primary research articles as well as appropriate organization of the material.
Line 47- should read "and have high recurrence rates"
Line 55- please cite primary research articles for each of the adverse outcomes listed instead of citing another review article
Add citation for the sentence Line 55-57 "Prolonged unresolved...."
Line 69- Citation number seven is about intracellular pathogens. Either add "intracellular" in the text, or add additional references to support the more general claim. Really the sentence should cite primary research articles and not another review article.
Line 76- "offering protection against disease as well as optimal conditions for pregnancy and childbirth" should be softened to "likely providing protection against disease and may also generally maintain optimal conditions for pregnancy and childbirth." It would be useful to provide examples of data indicating the protection against disease and clarify what is meant by optimal conditions for pregnancy and childbirth (with citations).
Line 77- "All these environments undergo significant shifts..." sentence needs a citation.
Line 91- "2.1 Urobiome" The overaching section title states "Reproductive Tract" not "Urogenital Tract"; including a urobiome section here doesn't match the section title. This paragraph is a fairly incomplete review of the the urobiome literature, missing several key citations. It is unclear what the authors goal is for including the urobiome section in this review. The main focus of this section appears to be on the relationship between the urobiome and BV. It is fine to have a focused section only on this aspect of the urobiome, but the title should indicate this and the text should give more indication that the section is not meant to review the full urobiome literature. The authors should provide more justification of why only these urobiome articles are being discussed, or, better yet, include discussion of other articles that have looked at the relationship between the urobiome and vaginal microbiome (again, if this section is intended to have that focus).
Line 111-117- This paragraph is unnecessary. It lacks citations and mainly suggests that longitudinal studies of the vaginal microbiome are largely lacking; they are not. There have been several longitudinal studies that have considered host and environmental factor influence. These should be discussed and cited. Indeed, some of them are in this very section.
Line 122- "This pH range also helps to inhibit many other invading microbes [14]." Citation 14 appears to be about vaginal acidification and not about pH inhibiting 'invading microbes.' Provide cition(s) for microbe inhibition.
Line 179- "Environments that are proximal to the gut (and therefore susceptible to bi-lateral spread from the gut) such as the peritoneal cavity and fallopian tubes contain almost no Lactobacillaceae and are highly diverse [24]." Citation 24 is inappropriate here (doesn't have any data about falopian tube microbiome). Include correct citation. There's no reason to mention the peritoneal cavity.
Line 204- "Additionally, bacteria can be introduced into the uterus via hematogenous spread from transmembrane gut leakage into the peritoneal cavity and through retrograde ascension from the fallopian tubes." Needs citation(s).
Line 210- "However, it is likely that resident microbiota plays a role..." recommend softening 'likely' to 'possible'.
Line 212- "Furthermore, the microbiome influences immunomodulatory factors that facilitate fetal development." Expand and provide citations
Line 216- It would have been much more helpful for this paragraph and citations to have appeared earlier in the article. It includes some of the citations justifying comments the authors make earlier in the article (that I have asked for above), and better sets up and frames Figure 1.
"Section 3.1 Bacterial vaginosis" is out of place. It should go up earlier in the article, in the section about taxonomic characterization. I recommend after section 2.2.
Line 257-258- "BV is caused by..." should be change to "BV is characterized by..."
Line 262- "three out of the four following criteria" should be "Amsel criteria" and should cite the original article instead of 42
Line 264- "BV occurs commonly in conjunction with infections..." needs more citations. The only thing cited is chronic endometritis.
Line 266- "There are also studies showing BV co-presenting with unexplained infertility..." needs more citations and the mention of the falopian tubes/ Gardnerella exotoxins result is weird. Why only mention this particular study (ref 44)? Also, reference 44 is in Bulgarian, so I cannot learn more about the Gardnerella exotoxin result, but the translated abstract makes no mention of it. Please clarify and provide more details, or remove this portion of the text.
Line 275- It is not clear what the authors mean with this sentence “Historically, disorders that affect female sexual health and quality of life have not garnered the necessary inquiry due to the potential sequelae of such a condition.” What do they mean by potential sequalae?
Line 279- “BV should be regarded as the most significant threat...” Change “the most significant” to “a significant”
Line 289- Vaginas are not pregnant; women are pregnant. Change to “vaginas of non-pregnant women”
Line 293- Glycogen deposits as chemotactic agents? Maybe glycogen breakdown products, but glycogen itself is a food source. Clarify this sentence.
Section 4.2 is not specific to pregnancy; therefore, it does not belong in the section about pregnancy and infertility.
Author Response
See attached word document. The authors thank the reviewer for their thorough and insightful critique.

Reviewer 2 Report
This is a well written review of the female urogential microbiome.
I think a reference to support the statement on lines 55-57 is needed.
Author Response
We added a reference for lines 55-57. We thank the reviewer for their comments.
Round 2
Reviewer 1 Report
Thank you to the authors for their efforts on the revised manuscript. It is much improved. I only caught a couple of minor typos.
Line 180- "Introduction" should not be capitalized
Line 325- "Of-tentimes" is incorrectly hyphenated.